# Cardiometabolic Risk after SARS-CoV-2 Virus Infection: A Retrospective Exploratory Analysis

**DOI:** 10.3390/jpm12111758

**Published:** 2022-10-24

**Authors:** Rute Pires, Miguel Pedrosa, Maria Marques, Margarida Goes, Henrique Oliveira, Hélder Godinho

**Affiliations:** 1Unidade de Cuidados Intensivos Polivalente (UCIP), Hospital de Espírito Santo de Évora EPE, 7000-811 Évora, Portugal; 2Unidade de Cuidados Intermédios Médicos (UCIM) e Área Respiratória do Serviço de Urgência Polivalente (AR-SUP), Hospital de Espírito Santo de Évora EPE, 7000-811 Évora, Portugal; 3Escola Superior de Enfermagem São João de Deus, Universidade de Évora, 7004-516 Évora, Portugal; 4Comprehensive Health Research Centre (CHRC), Universidade de Évora, 7004-516 Évora, Portugal; 5Instituto de Telecomunicações (IT-Lisboa), 1049-001 Lisboa, Portugal; 6Instituto Politécnico de Beja, 7800-295 Beja, Portugal

**Keywords:** cardiometabolic risk, SARS-CoV-2 infection, intensive care unit

## Abstract

Objective: The aim of this study is to characterize the cardiometabolic risk of individuals who were infected with the SARS-CoV-2 virus and subsequently admitted to a hospital in a major city in mainland Portugal. Methods: This is a retrospective exploratory study using a sample of 102 patients, with data analysis including descriptive statistics, nonparametric measures of association between variables based on Spearman’s rank-order correlation, a logistic regression model for predicting the likelihood that an individual might eventually pass away, and a multiple linear regression model to predict a likely increase in the number of days an infected patient remained in the hospital. Results: About 62.7% of the individuals required intensive care on the second day of hospitalization, remaining 14.2 days in the intensive care unit (ICU) on average. The likelihood that an individual might eventually pass away due to SARS-CoV-2 virus infection increases for the older than younger ones and increases even more if he/she suffers from cardiometabolic disorders such as obesity, especially cardiovascular disease. Older individuals and those with obesity and hypertension remained more days in the ICU. Conclusions: A later age and the prevalence of cardiometabolic disorders severely affect the care pathway of individuals infected with the SARS-CoV-2 virus.

## 1. Introduction

Cardiometabolic disorders represent a cluster of interrelated risk factors for an individual’s health: hypertension, type 2 diabetes, dyslipidemia, abdominal obesity, and cardiovascular disease. They aggravate the individual’s clinical condition mainly when they co-occur [1]. The acute respiratory syndrome coronavirus-2 (SARS-CoV-2) infection, also known as coronavirus disease 2019 (COVID-19), has brought severe health consequences for many of its carriers [2], especially for those with cardiometabolic disorders [3,4].

Diabetes is a risk factor that influences the progression and prognosis of COVID-19, commonly associated with high mortality rates among patients with acute chronic disorders such as cardiovascular disease. A recent study reported that elevated fasting blood sugar levels in SARS-CoV-2 infected patients were strongly correlated with mortality, whether or not they had diabetes [5]. Hyperglycemia is an independent predictor of ineffective breathing that increases the risk of respiratory tract infection, i.e., it is an indicator of a poor prognosis regarding patient recovery [4]. According to Barron et al., diabetes and/or uncontrolled acute hyperglycemia, associated with an increase in the number of days a patient remained in the hospital, increase the mortality rate due to COVID-19 [3].

Obesity, especially abdominal obesity, is considered a chronic disease and an important trigger for the development of diabetes and many of its associated conditions. Moreover, the combined harmful effects of obesity and type 2 diabetes affect the human body’s tissues, resulting in a significant increase in premature morbidity and mortality [6].

Cardiometabolic disorders increase the chance of SARS-CoV-2 infected patients developing acute cardiovascular changes, especially arrhythmias, cardiac arrest, acute myocardial infarction, and heart failure, as reported in recently published research [7]. However, these acute changes can lead to chronic cardiovascular damage or even death, even for those without cardiovascular disorders prior to SARS-CoV-2 infection [8].

All of the metabolic diseases, including excess weight, obesity, metabolic syndrome, type 2 diabetes, and vascular diseases, have achieved epidemic status worldwide [9,10,11,12,13,14,15], with cardiometabolic disorder representing an additional risk factor for the development of cardiovascular disease, among other health disorders, before and after SARS-CoV-2 infection [16].

It is estimated that approximately 25% of hospitalized patients with COVID-19 have cardiometabolic disorders [17]. Thus, given the evidence presented, the study of cardiometabolic disorders associated with SARS-CoV-2 infection in patients admitted to an intensive care unit (ICU) will contribute to comprehending the patient pathway (or care pathway) from the time he/she was admitted to the hospital until he/she no longer needed to receive inpatient care and could go home (hospital discharge).

## 2. Materials and Methods

The Health Ethics Committee of the “Hospital do Espírito Santo de Évora, EPE” (CES HESE, EPE [18]) approved the study protocol on 7 July 2021. The decision was published in the meeting minutes issued by the CES HESE, EPE board of directors, with reference number 28. In addition, all the research methods were performed in full compliance with the statements included in the operating regulations of the CES HESE, EPE [18], a document that was developed in accordance with the Helsinki Declaration, aiming to protect the dignity, privacy, and freedom of the participants.

This retrospective exploratory study involved individuals infected with the virus that causes COVID-19 disease who were assisted in a hospital located in a major city in mainland Portugal from 25 March 2020 to 27 February 2021. All participants were registered in the database of the “Unidade de Cuidados Intensivos Polivalente do Hospital do Espírito Santo de Évora, EPE”, comprising 102 individuals who were not vaccinated against COVID-19 disease. Patient consent was waived since this was a retrospective study containing data from patients who had already passed away, with full assurance of the anonymity and confidentiality of all data relating to all subjects.

The variables considered in this research, in addition to *sex* and *age* of patients, were: (i) the number of days between the dates of a positive COVID-19 test result and hospitalization (scale variable); (ii) the number of days between the dates of hospitalization and ICU admission (scale variable); (iii) the number of days an individual remained in the ICU (scale variable); (iv) the cardiometabolic disorders before an individual remained in the hospital (either in an inpatient area or the ICU) such as *obesity*, *diabetes*, *dyslipidemia*, *hypertension*, and *cardiovascular disease*—CVD (nominal variables, one per each disorder, whose answers were “Yes” and “No”); and (iv) whether he/she passed away (a nominal variable, whose answers were “Yes” and “No”).

To assess the significance of *age*, *sex*, as well as the five cardiometabolic disorders considered within the scope of this study in predicting the likelihood that an individual might eventually pass away during the days he/she remained in the hospital (either in an inpatient area or the ICU), a logistic regression model using the *ENTER* method, and then the *Forward:LR* method, as described in the study by Marôco [19], was used.

Multiple linear regression with a *stepwise* selection of variables was used to develop three parsimonious models aiming to identify which independent variables (*sex*, *age*, *obesity*, *diabetes*, *dyslipidemia*, *hypertension*, and *CVD*) were the best predictors of: (i) a likely increase in the number of days between the dates of a positive COVID-19 test result and hospitalization (inpatient admission); (ii) a likely increase in the number of days an infected patient remained in the inpatient area of the hospital; and (iii) a likely increase in the number of days an infected patient remained in the ICU.

All statistical analyses were performed with the software IBM SPSS Statistics for Windows v.27 (IBM Corp., Armonk, NY, USA). Type I error probabilities (*α*) of 0.05 and 0.10 were considered for all analyses.

## 3. Results

### 3.1. Descriptive Analysis

Table 1 lists the results of the descriptive analysis. The cohort used in this research comprised 102 individuals, of whom 67.6% were male (the average age was 66.0, ranging from 27 to 92 years) and 32.4% were female (the average age was 68.3, ranging from 42 to 94 years). More than half of the individuals presented multimorbidity (61.4%), while 16.9% had only one cardiometabolic disorder, with almost one-quarter of the cohort comprising healthy individuals. The most common cardiometabolic disorders among individuals were *hypertension* (55.4%), *dyslipidemia* (34.7%), *diabetes* (29.7%), *obesity* (25.7%), and CVD (23.8%). Of the 102 individuals in the cohort, 101 were admitted to the ICU, of whom 28 passed away. No one passed away in the inpatient area of the hospital.

The number of days between the dates of a positive COVID-19 test result and hospitalization (inpatient admission) was counted. The average was 4.1 days, ranging from 0 (minimum) to 13 (maximum) days, although about one-quarter of the individuals (26.1%) were hospitalized on the same day they tested positive for COVID-19 infection (i.e., 0 days). In addition, the average number of days between the dates of hospitalization and ICU admission was 2.9, ranging from 0 (minimum) to 24 (maximum), although more than half of the subjects (62.7%) had required intensive care by the second day of hospitalization. Finally, the average number of days an individual remained in the ICU was 14.2 (value for the entire cohort), ranging from 0 (minimum) to 53 days (maximum) and from 1 (minimum) to 85 days (maximum) for those who passed away and those who survived, respectively.

### 3.2. Nonparametric Measures of Associations

Table 2 lists the results regarding nonparametric measures of associations between some variables considered within the scope of this study using Spearman’s rank-order correlation. They seem to suggest that the number of days between the dates of a positive COVID-19 test result and hospitalization (inpatient admission) was associated with the individuals’ age (0.21, qualitatively rated as weak and statistically significant at 0.10 level) but not with the total number of their cardiometabolic disorders. Furthermore, the number of days between the hospitalization and ICU admission dates appears to be positively associated only with the individuals’ age (0.19, qualitatively rated as weak and statistically significant at 0.10 level). Finally, the associations between the number of days he/she remained in the ICU and the total number of cardiometabolic disorders with the individuals’ age were statistically highly significant (0.30, qualitatively rated as weak) and significant (0.22, qualitatively rated as weak), respectively.

### 3.3. Logistic Regression Model

The logistic regression model assumptions were validated by graphical analysis of the residuals, and the existence of outliers was diagnosed. Only one case was considered an outlier, but it was incorporated into the model since its removal did not improve the statistical significance of the logistic model or its goodness-of-fit. The logistic regression model with all its predictors, using the *Enter* selection method, revealed that *sex* and the various cardiometabolic disorders (except *obesity* and *CVD*) had no statistically significant effect on the *Logit* of the likelihood that an individual might eventually pass away due to SARS-CoV-2 virus infection. On the contrary, the variables *age*, *obesity*, and *CVD* had a statistically significant effect on the *Logit* of the likelihood that an individual might eventually pass away due to SARS-CoV-2 virus infection. Table 3 summarizes the model coefficients and their statistical significance.

Thus, a new statistically significant model was fitted employing the *Forward: LR* method $\left(G^{2}\left(3\right)=9.973;p=0.019;\chi_{HL}^{2}\left(7\right)=7.880;p=0.343;R_{CS}^{2}=0.119;R_{N}^{2}=0.171;R_{MF}^{2}=0.107\right)$, using the following variables: (i) CVD $\left(B_{CVD}=0.893;\chi_{Wald}^{2}\left(1\right)=2.787;p=0.095;OR=2.442\right)$; (ii) obesity $\left(B_{Obesity}=0.221;\chi_{Wald}^{2}\left(1\right)=2.912;p=0.088;OR=1.247\right)$; and (iii) age $\left(B_{age}=0.051;\chi_{Wald}^{2}\left(1\right)=4.404;p=0.036;OR=1.052\right)$.

The probability function that an individual might eventually pass away after being infected with the SARS-CoV-2 virus and subsequently hospitalized, as a function of *age*, *obesity*, and *CVD*, is shown in Figure 1. The corresponding models, statistically significant at 0.10 level, are:(1)$$\hat{\pi }\left(CVD_{“Yes”}\&Obesity_{“Yes”}\right)=\frac{1}{1+e^{-\left[-3.763+0.893+0.221+0.051\times age\right]}}$$
(2)$$\hat{\pi }\left(CVD_{“No”}\&Obesity_{“Yes”}\right)=\frac{1}{1+e^{-\left[-3.763+0.221+0.051\times age\right]}}$$
(3)$$\hat{\pi }\left(CVD_{“Yes”}\&Obesity_{“No”}\right)=\frac{1}{1+e^{-\left[-3.763+0.893+0.051\times age\right]}}$$
(4)$$\hat{\pi }\left(CVD_{“No”}\&Obesity_{“No”}\right)=\frac{1}{1+e^{-\left[-3.763+0.051\times age\right]}}$$

The model correctly classified about 74.7% of cases, which is considered reasonably higher than the proportional percentage of correct random classifications (52%), reasonably demonstrating its usefulness for classifying new cases. The model’s sensitivity (true positive rate—TPR) was higher (93.0%), although the specificity (false predicting rate—FPR) was poor (only 27.3%), and the discriminating ability was reasonable (AUC = 0.733, *p* = 0.001, see Figure 2). The overall value of the Youden index was *J* = 0.203. Regarding the variable *age*, the maximum value found for *J* was 0.348 (a weak value) for 68.5 years old, which is considered the optimal cutoff value to predict loss of life among hospitalized individuals.

### 3.4. Multilinear Regression Models

The models’ assumptions were analyzed: normal distribution, homogeneity, and error independence. The first two assumptions were validated graphically, and the assumption of the independence of errors was validated with the Durbin–Watson statistic (d = 1.66), as described in the study by Marôco [19]. The variance inflation factor (VIF) was used to diagnose multicollinearity among the variables, which did not occur. The diagnosis of outliers was carried out (data with a studentized residual, in absolute value, greater than 1.96), although only two outlier cases were found that were not removed from the models since they did not significantly affect their goodness-of-fit.

The first multiple linear regression model with a *stepwise* selection of variables was designed using the number of days between the dates of a positive COVID-19 test result and hospitalization (inpatient admission) as the dependent variable and the variables *sex*, *age*, and the five cardiometabolic disorders as dependent variables. This model $\left(F\left(1,90\right)=4.264;p=0.042;\;R_{a}^{2}=0.243\right)$ allowed only the variable *sex* (B=−0.213;t=−2.065; p=0.042) to be identified as a statistically significant predictor of the dependent variable (see results in Table 4).

The second multiple linear regression model with a *stepwise* selection of variables was designed using the number of days between the dates of hospitalization (inpatient admission) and ICU admission as the dependent variable and the variables *sex*, *age*, and the five cardiometabolic disorders as dependent variables. This model $\left(F\left(1,97\right)=4.306;p=0.041;\;R_{a}^{2}=0.206\right)$ allowed only *obesity* (B=−0.206;t=−2.075; p=0.041) to be identified as a statistically significant predictor of the dependent variable (see results in Table 5).

Finally, a third multiple linear regression model with a *stepwise* selection of variables was designed using the number of days an individual remained in the ICU as the dependent variable and the variables *sex*, *age*, and the five cardiometabolic disorders as dependent variables. This model, statistically significant at 0.10 level $\left(F\left(2,75\right)=2.946;p=0.059;\;R_{a}^{2}=0.270\right)$, allowed *age* (B=0.236;t=2.119; p=0.037) and *hypertension* (B=0.220;t=1.752; p=0.084) to be identified as statistically significant predictors of the dependent variable (see results in Table 6).

## 4. Discussion

The cohort considered in this retrospective exploratory study included only individuals admitted to the hospital of a large city in mainland Portugal after being infected with the virus that causes COVID-19 disease. The majority (approximately two-thirds) were men, which aligns with findings already reported by other researchers. For example, according to Medzikovic et al., men infected with the virus that causes COVID-19 appeared to suffer worse disease progression and higher mortality rate than women due to the complex interplay between biological and societal factors, along with the prevalence of cardiometabolic disorders such as CVD [20].

More than half of the individuals of the cohort had multimorbidity (61.4%), a clinical condition defined as the co-occurrence of two or more chronic conditions, which is considered a relevant public health concern in Portugal [21,22]. For example, a National Health Service (NHS) report revealed that in 2016, 41% of the total Portuguese population had multimorbidity (11% had two chronic diseases, 8% had three, and 22% had four or more chronic diseases), with 18% already having one chronic disease [23]. Additionally, the number of cardiometabolic disorders was found to be significantly associated with the individuals’ age for the cohort considered in this article (qualitatively rated as “weak”), which is in line with the fact that multimorbidity increases with *age*, as also stated in the NHS report, becoming more prevalent in women than men [23]. Furthermore, according to the research published by Rodrigues et al. in 2019 concerning the first National Health Examination Survey, the authors reported a high prevalence of multimorbidity among individuals aged 65 and older, estimated at 78.3% for the Portuguese population sample used in their research (estimates reported by age groups of 65–69, 70–74, 75–79, and 80 years and older: 72.8%, 78.2%, 81.9%, and 83.4%, respectively) [24].

According to published research, older people and those with underlying medical problems, such as CVD, diabetes, chronic respiratory disease, and cancer, are at a higher risk of developing severe COVID-19 illness [25,26,27]. For the cohort considered in this article, the number of days that the individuals remained in the ICU was not significantly associated with their number of cardiometabolic disorders. The same outcome was obtained with respect to the number of days between the dates of a positive COVID-19 test result and hospitalization (inpatient admission) and between the dates of hospitalization and ICU admission. On the contrary, the individuals’ age was found to be significantly associated (qualitatively rated as “weak”) with the number of days that they remained in the ICU, i.e., older individuals remained more days in the ICU than younger ones. As is well-known, older individuals have a lower functional reserve of organs and organ systems than younger individuals. In addition, aging is associated with chronic inflammation and oxidative stress, increasing CVD risk, along with new age-related cardiovascular risk factors, such as frailty and sarcopenia, which are emerging [28]. All of this may contribute to the prognosis of the increasing average number of days he/she remained in the ICU [29].

As for the results obtained by the logistic regression model, the likelihood that a person might eventually pass away after being infected with the SARS-CoV-2 virus and later being hospitalized either in an inpatient area or in the ICU increases exponentially with age, i.e., the chance of eventually passing away versus not passing away increases by 5.2% for each year of the individual’s age. The age of 68.5 years old was considered the optimal cutoff value to predict loss of life among hospitalized individuals in the ICU. For a given age, the likelihood that an individual might eventually pass away is always higher if he/she has obesity, i.e., the chance of eventually passing away versus not passing away increases by 24.7% if he/she has obesity versus not. On the other hand, should an individual suffer from CVD, the likelihood that he/she might eventually pass away is even higher, i.e., the chance of eventually passing away versus not passing away increases by 1787% if he/she has CVD versus not. Finally, no inference could be made regarding the effect of the remaining cardiometabolic disorders, such as diabetes, hypertension, and dyslipidemia, because the corresponding variables were not statistically significant in the logistic regression model. However, diabetes had the highest odds ratio among these three about whether an individual might eventually pass away. Several studies conducted since the beginning of the pandemic have shown that obesity is a risk factor for developing severe complications in case of SARS-CoV-2 infection and, consequently, with increased risk of death from the infection, just as it increases the risk of hospitalization and need for intensive care [30].

Concerning the different multiple linear regression models designed for each of the three dependent variables, the following statistically significant predictors were identified: (i) sex (a negative effect of −0.213 in the case of men compared to women), with respect to the number of days between the dates of a positive COVID-19 test result and hospitalization (inpatient admission); (ii) obesity (a negative effect of −0.206), with respect to the number of days between the dates of hospitalization and ICU admission; and (iii) age (a positive effect of 0.236) and hypertension (a positive effect of 0.220, statistically significant at 0.10 level), with respect to the number of days an individual remained in the ICU. Based on these results, it can be inferred that after finding out they were infected with the SARS-CoV-2 virus through a positive COVID-19 test result, women took more days to visit the hospital and consequently were hospitalized later than men. This finding may be related to women’s generally more robust response to infectious pathogens [31,32] and a greater likelihood of adopting non-pharmacological measures to prevent infection, mainly hand hygiene [33] and preventive behaviors [34]. In addition, individuals with obesity remained less time in the inpatient area of the hospital and were consequently admitted to the ICU in a shorter time, which is a finding that is aligned with those published by other researchers that associates obesity with a worse progression of COVID-19 disease [35], namely the prediction of a higher risk of infection [36], quicker ICU admission [37], the need for invasive mechanical ventilation [38], and a higher mortality rate [39]. Finally, older individuals and those with hypertension remained in the ICU longer than others. It is well-known that the mortality rate from SARS-CoV-2 virus infection is higher among older adults than younger ones. However, more men have been dying than women, which can be explained by the higher burden of chronic diseases in this population group, such as diabetes, smoking habits, or alcohol consumption [40]. Similarly, Ribeiro [41] concluded that males, those of an older age group, and being hospitalized are associated with a higher risk of death from SARS-CoV-2 virus infection. Recent studies also highlight some predictive risk factors for SARS-CoV-2 virus infection in the population that consequently lead to severe clinical outcomes of COVID-19 disease, namely: advanced age; chronic lung disease; immunocompromised status; and the cumulative prevalence of comorbidities such as hypertension, diabetes, or CVD [42].

## 5. Conclusions

In this paper, the cardiometabolic disorder prevalence’s effect on the care pathway of the SARS-CoV-2 virus-infected patient, in terms of: (i) death; (ii) the number of days between the dates of a positive COVID-19 test result and hospitalization (inpatient admission); (iii) the number of days an infected patient remained in the inpatient area of the hospital; and (iv) the number of days an infected patient remained in the ICU, was investigated. It was found that cardiometabolic disorders promote a worse COVID-19 disease progression, resulting with some frequency in the need for care assistance in the ICU. Therefore, individuals with a higher degree of cardiometabolic disorders present an increased risk of developing adverse outcomes after being infected by the SARS-CoV-2 virus.

To further understand the cardiometabolic disorder prevalence’s effect on the care pathway of the SARS-CoV-2 virus infected patient from the time he/she was admitted to the hospital until he/she no longer required hospital care and could go home (hospital discharge), further studies are needed in other health units, either in Portugal or other countries. Moreover, rehabilitation initiated in the ICU should continue throughout the hospitalization and after patient discharge through multidisciplinary outpatient care, home health services, and peer support groups, which is a topic that deserves to be investigated, as suggested by some researchers.

Like all scientific studies, this one also has its limitations, in which the most significant ones are listed as follows: (i) the low number of individuals in the cohort, which could affect the quality of the results obtained by inference methods; (ii) the lack of data on the type of treatment each patient underwent, which could have affected the results; and (iii) since this study was conducted only in a hospital in a large city in mainland Portugal, the results cannot be generalized to other healthcare facilities in the country.

## Figures and Tables

**Figure 1 jpm-12-01758-f001:**
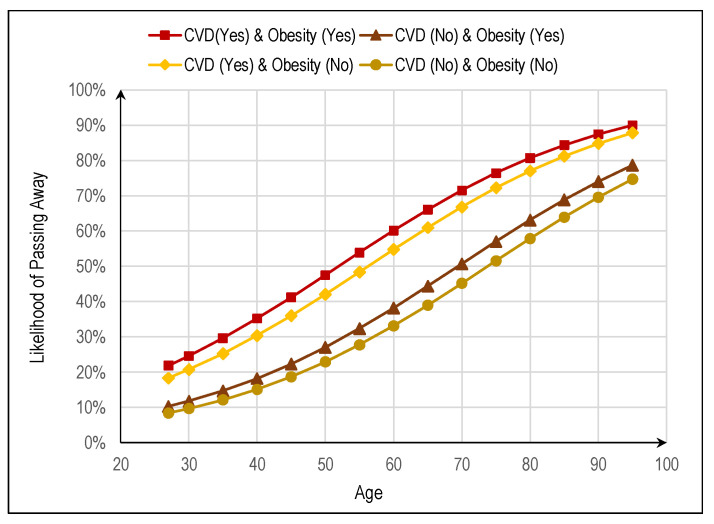
The likelihood that an individual might eventually pass away as a function of *age*, *obesity*, and *CVD*, with the models being statistically significant at 0.10 level.

**Figure 2 jpm-12-01758-f002:**
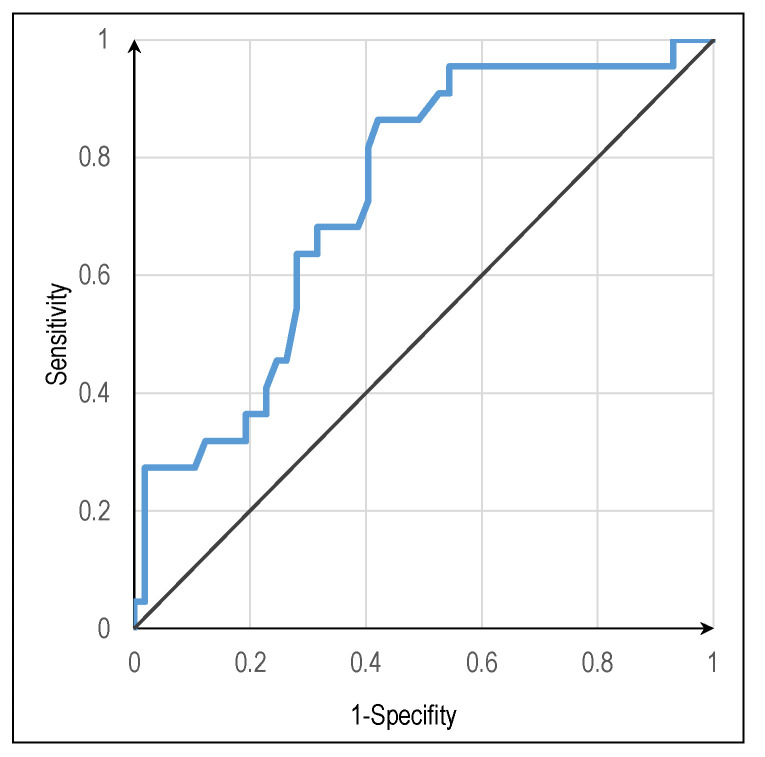
Receiver operating characteristic (ROC) curve showing the trade-off between sensitivity (or TPR) and specificity (1—FPR). Area under the curve (AUC) was 0.733 (*p* = 0.001).

**Table 1 jpm-12-01758-t001:** Descriptive analysis.

Variables	N	%	Min/Max/Avg
Sex:			
Male	33	32.4	
Female	69	67.6	
Age:			
Male	-	-	27/92/66.0
Female	-	-	42/94/68.3
Cardiometabolic disorders:			
None	18	21.7	
One	14	16.9	
Two	19	22.9	
Three	22	26.5	
Four	10	12.0	
Hypertension	56	55.4	
Dyslipidemia	35	34.7	
Diabetes	30	29.7	
Obesity	26	25.7	
CVD	24	23.8	
Days between dates of a positive COVID-19 test and hospitalization	-	-	0/13/4.1
Days between dates of hospitalization and ICU admission	-	-	0/24/2.9
Days in ICU:	-	-	0/85/14.2
Passed away	-	-	0/53/16.2
Survived	-	-	1/85/13.5

**Table 2 jpm-12-01758-t002:** Nonparametric measures of association between variables.

Variables	Age	Total Number of Cardiometabolic Disorders
Days between positive COVID-19 test and hospitalization	0.21 (*p* = 0.051) ^1^	0.03 (*p* = 0.777)
Days between hospitalization and ICU admissions	0.19 (*p* = 0.063) ^1^	0.09 (*p* = 0.389)
Days in ICU	0.30 (*p* = 0.007)	0.04 (*p* = 0.711)
Total number of Cardiometabolic disorders	0.22 (*p* = 0.033)	-

^1^ Significant at 0.1 level.

**Table 3 jpm-12-01758-t003:** Results of logistic regression model using the *Enter* selection method.

Variables	*B* ^1^	S.E. ^2^	$\chi_{Wald}^{2}$ ^3^	*d.f.* ^4^	*p*	Exp(B) ^5^	95% CI for Exp(B) ^6^
Sex	0.163	0.663	0.061	1	0.806	1.177	[0.321; 4.318]
Age	0.046	0.026	4.017	1	0.045	1.047	[1.003; 1.098]
Obesity	0.338	0.755	3.039	1	0.081	1.402	[0.978; 4.686]
Diabetes	0.803	0.604	1.769	1	0.183	2.232	[0.684; 7.288]
Dyslipidemia	0.142	0.604	0.155	1	0.694	1.153	[0.266 3.835]
Hypertension	0.052	0.607	0.007	1	0.932	1.053	[0.289; 3.120]
CVD	0.988	0.623	2.767	1	0.096	2.686	[0.816; 9.939]
Constant	−3.763	1.809	4.326	1	0.038	0.023	-

^1^*B*: Regression coefficient; ^2^ S.E.: Standard error; ^3^ $\chi_{Wald}^{2}$: the Wald test statistics; ^4^
*d.f.*: Degrees of freedom; ^5^ Exp(B): Exponential of B, the odds ratio (OR); ^6^ CI: Confidence interval.

**Table 4 jpm-12-01758-t004:** Results of multilinear regression model (*stepwise*) that was designed using the number of days between the dates of a positive COVID-19 test result and hospitalization (inpatient admission) as the dependent variable.

Independent Variables	*B* ^1^	S.E. ^2^	Standardized *B* ^3^	*t*	*p*
Sex	−1.898	0.919	−0.213	−2.065	0.042
Age	0.038	0.038	0.108	1.000	0.320
Obesity	−1.302	0.993	−0.140	−1.312	0.193
Diabetes	−1.301	0.946	−0.144	−1.375	0.173
Dyslipidemia	−0.668	1.006	−0.078	−0.664	0.508
Hypertension	1.098	0.885	0.131	1.240	0.218
CVD	0.719	1.042	0.076	0.690	0.492
Constant	5.894	0.878	-	6.709	<0.001

^1^*B*: Regression coefficient; ^2^ S.E.: Standard error; ^3^ Standardized regression coefficient.

**Table 5 jpm-12-01758-t005:** Results of multilinear regression model (*stepwise*) that was designed using the number of days between the dates of hospitalization (inpatient admission) and ICU admission as the dependent variable.

Independent Variables	*B* ^1^	S.E. ^2^	Standardized *B* ^3^	*t*	*p*
Sex	0.291	0.928	0.034	0.313	0.755
Age	0.009	0.039	0.026	0.227	0.821
Obesity	−1.875	0.904	−0.206	−2.075	0.041
Diabetes	0.604	0.902	0.069	0.670	0.505
Dyslipidemia	1.008	0.903	0.120	1.116	0.267
Hypertension	0.038	0.910	0.005	0.041	0.967
CVD	0.316	1.008	0.034	0.314	0.754
Constant	2.919	0.568	-	5.137	<0.001

^1^*B*: Regression coefficient; ^2^ S.E.: Standard error; ^3^ Standardized regression coefficient.

**Table 6 jpm-12-01758-t006:** Results of multilinear regression model (*stepwise*) that was designed using the number of days an individual remained in the ICU as the dependent variable.

Independent Variables	*B* ^1^	S.E. ^2^	Standardized *B* ^3^	*t*	*p*
Sex	−0.401	3.556	−0.014	−0.113	0.911
Age	0.277	0.131	0.236	2.119	0.037
Obesity	3.310	3.783	0.107	0.875	0.385
Diabetes	4.535	3.503	0.151	1.295	0.200
Dyslipidemia	1.863	3.504	0.065	0.532	0.597
Hypertension	6.055	3.457	0.220	1.752	0.084 ^4^
CVD	−4.548	3.809	−0.142	−1.194	0.237
Constant	−3.889	8.726	-	−6.575	<0.001

^1^*B*: Regression coefficient; ^2^ S.E.: Standard error; ^3^ Standardized regression coefficient; ^4^ Statistically significant at 0.10 level.

## Data Availability

Not applicable.

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
