# Peer review of "Cardiometabolic Risk after SARS-CoV-2 Virus Infection: A Retrospective Exploratory Analysis"

_jpm, 2022, doi:10.3390/jpm12111758_

Round 1
Reviewer 1 Report
I think that this manuscript was well done and well written. I don't have any suggestion to do.
Congratulations all the authors.
Author Response
Reviewer 1 – General Comments:
I think that this manuscript was well done and well written. I don't have any suggestion to
do. Congratulations all the authors.
Response: (RP+MP+MM+MG+HO+HG)
The authors would like to thank the Reviewer for the very kind words regarding our
proposal. All authors believe the proposal has some scientific background to be considered
for publication in a journal like the “Journal of Personalized Medicine” of MDPI.
Reviewer 2 Report
See the attached file

Author Response
Reviewer 2 – General Comments:
The article is well written and the goal has been clearly stated. However, some
methodological issues should be better addressed (see below). Moreover, as the results
are quite expected based on the existing literature, the authors may perhaps try to better
present the relevance of their findings, by further delving into the links with the existing
literature for instance.
Response: (RP+MP+MM+MG+HO+HG)
The authors would like to thank the Reviewer for the very kind words regarding our
proposal. All authors believe the proposal has some scientific background to be considered
for publication in a journal like the “Journal of Personalized Medicine” of MDPI. However, all
authors fully agree that the proposal needs to be updated to clearly address the essential and
highly relevant issues mentioned by the Reviewer.
Major comments: 1
Table 1: one of the outcomes is death during a subject remained in the hospital, however
the number of deaths has not been indicated. The same for the number of subjects
admitted to ICU. This is of general interest and particularly relevant in logistic regression
analysis, as the suitable number of covariates depends on the number of successes and
failures (a quite known rule of thumb for logistic regression is that you need 10-20 cases
in the lower-prevalence category per parameter whose value you are trying to estimate).
Response: (RP+MP+MM+MG+HO+HG)
Thank you for pointing out this critical aspect. Unfortunately, the number of deaths and
individuals admitted to the ICU were not indicated in the proposal, either in text or tables. It
was an oversight by the authors, who apologize for the fact. The cohort comprised
individuals assisted in the hospital from March 25th, 2020, until February 27th, 2021. Of the
102 individuals registered in the hospital database during this period, 101 were admitted to
the ICU, of whom 28 passed away. No one passed away in the inpatient area of the hospital.
Regarding the minimal number of events per variable, some authors study the importance
and validity of the mentioned rule of thumb, like van Smeden et al. (van Smeden et al., 2016).
In their article, in conclusions, the authors state that the evidence underlying the ten-events-
per-variable rule as a minimum sample size criterion for binary logistic regression analysis
is weak. They also mention that Firth’s correction (which was not considered in the
proposal) can be used to improve the accuracy of regression coefficients and alleviate the
problems regarding the low numbers of events per variable associated with separation.
However, this is an undergoing field of research, and more studies are urgently needed to
provide guidance for supporting sample size considerations for binary logistic regression
analysis. Nevertheless, concerning the well-known logistic regression rule mentioned by the
reviewer, the proposal's authors humbly believe that the sample considered does not

Round 2
Reviewer 2 Report
I thank the authors for the careful and thorough revision of the manuscript, as well as for the detailed responses they have provided to my comments. I very much appreciated the rigor and clarity of the work: my congratulations.